# Imagined Autocurricula

**Ahmet H. Güzel**[*]
University College London AI Centre

**Matthew T. Jackson**
University of Oxford

**Jarek L. Liesen**
University of Oxford

**Tim Rocktäschel**
University College London AI Centre

**Jakob N. Foerster**
University of Oxford

**Ilija Bogunovic**
University College London AI Centre

**Jack Parker-Holder**
University College London AI Centre

## Abstract

Training agents to act in embodied environments typically requires vast training data or access to accurate simulation, neither of which exists for many cases in the real world. Instead, *world models* are emerging as an alternative–leveraging offline, passively collected data, they make it possible to generate diverse worlds for training agents in simulation. In this work, we harness world models to generate "imagined" environments to train robust agents capable of generalizing to novel task variations. One of the challenges in doing this is ensuring the agent trains on *useful* generated data. We thus propose a novel approach IMAC (Imagined Autocurricula) leveraging Unsupervised Environment Design (UED), which induces an automatic curriculum over generated worlds. In a series of challenging, procedurally generated environments, we show it is possible to achieve strong transfer performance on held-out environments having trained only inside a world model learned from a narrower dataset. We believe this opens the path to utilizing larger-scale, foundation world models for generally capable agents.

## 1 Introduction

Despite significant advancements in AI, we remain far from generally capable agents [1]. In the 2010s, progress towards this goal was driven by breakthroughs in deep reinforcement learning (RL, [2]) where agents such as AlphaGo [3] showed it was possible to discover new knowledge beyond human capabilities, like the famous *move37*. However, scaling RL in simulation to produce a more general intelligence remains bottlenecked by the lack of sufficiently diverse and real-world simulators [4, 5]. By contrast, more recent years have been dominated by agents trained from vast quantities of Internet data. Here, agents have a broader knowledge of the real world but lack of ability to discover new, superhuman behaviors since they are largely trained to mimic the training data.

World models are emerging as a promising approach to leverage the best of both. They "stand on the shoulders of giants" by training on vast Internet data [6, 7, 8, 9, 10], but then enable agents to train with RL inside their generated (or *imagined*) environments. Thus, world models make it possible for agents to imagine potential outcomes without direct environment interaction, allowing them to train across diverse scenarios with substantially fewer real experiences [11, 12, 13, 14, 15, 16, 17]. In theory, large foundation world models could then be used to generate sufficiently rich environments for agents to learn superhuman behaviors for a swathe of embodied settings. Progress in world models research has typically fallen into two categories.

---

[*]Corresponding author (ahmet.guzel.23@ucl.ac.uk).

39th Conference on Neural Information Processing Systems (NeurIPS 2025).

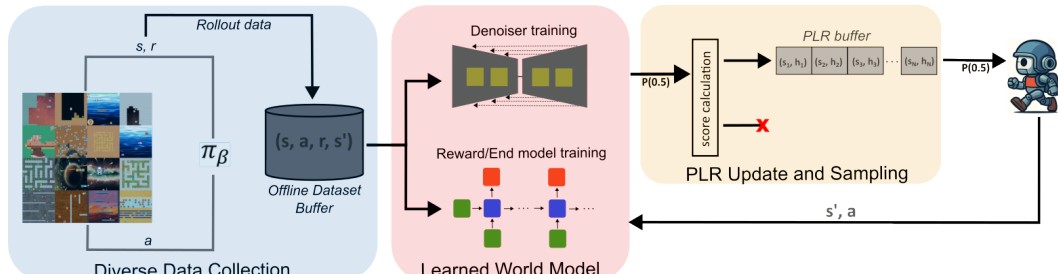

Figure 1: Architecture overview: (1) Offline Data Collection: Diverse state-action-reward-next state tuples are collected via policy $\pi_\beta$ to form the dataset. (2) World Model Training: A state transition denoiser and reward/termination predictor are trained on the dataset and frozen for consistency. (3) Agent Training: The agent learns through an imagine-based autocurriculum using Prioritized Level Replay, which balances between sampling from existing experiences and generating new rollouts to update the buffer.

The first seeks to train world models from single (online or offline) environments, achieving strong and sample-efficient performance [18, 19, 17]. The latter sees more general, so-called foundation world models trained from large-scale internet data [6, 7, 9, 10], but these models have not been shown to be useful for training generalist agents.

In this paper, we attempt to bridge this gap. We focus on the offline RL setting, where agents must learn from pre-collected datasets without additional environment interaction [20]. We focus on procedurally generated environments, meaning our offline data contains sequences (action labelled videos) from a variety of different levels. We then seek to transfer the learned agents to new, unseen levels from the same environment. This presents a significant challenge for existing methods, which often fail to generalize to new tasks when training purely from offline data [21, 22]. We believe world models can be the solution to this problem–we first train a diffusion world model from the offline samples and then use it to generate new, *imagined* worlds. Crucially, naively generating data from world models can lead to ineffective training data, making agent learning inefficient. This inefficiency stems from both the quality of the worlds themselves and the challenge of setting appropriate hyperparameters like imagined episode length across diverse tasks. To address this challenge, we leverage the Unsupervised Environment Design paradigm (UED) [23]—where a teacher proposes levels that maximize student regret. Rather than training on fixed or random sets of imagined worlds, we create an Imagined Autocurricula, or IMAC. Importantly, our framework is world-model agnostic—while we use a diffusion world model as our foundation, our autocurriculum approach can be applied to any world model architecture that provides reward and next-token predictions. Our contribution lies not in advancing world model architectures, but in demonstrating how UED principles can effectively guide agent training within learned world models from offline mixed dataset.

Our approach IMAC uses Prioritized Level Replay (PLR, [24]) as a UED algorithm, which we show provides a natural complement to the learned world model–the world model generates diverse potential training trajectories or "imagined environments," while PLR strategically selects subsequent training tasks from these imagined rollouts. Figure 1 illustrates the overall architecture of our approach. As we show, this prioritization process naturally induces an automatic curriculum over the generated, imagined worlds, meaning the agent is exposed to increasingly challenging training tasks. As a testbed for this paradigm, we leverage seven Procgen environments with high-dimensional visual inputs, where both model-free [25] and model-based approaches [26] struggle to transfer knowledge to unseen scenarios [21, 22]. The resulting agents have significantly stronger generalization performance than state-of-the-art baselines, demonstrating strong transfer performance on held-out environments having trained only inside a world model learned from a narrower dataset. We believe this opens the path to utilizing larger-scale, foundation world models for generally capable agents.

Our primary contribution is the demonstration that world models offer the path to training on diverse offline datasets and subsequently generalizing to new, unseen tasks. Importantly, this is made possible by training on an Imagined Autocurriculum; rather than fixing hyperparameters or randomizing settings, each of which could be problematic for learning. We believe this is the first example of open-ended learning in learned world models, and could enable significant progress towards generalist agents with ever more powerful world models in the future.

## 2 Preliminaries

### 2.1 (Offline) Reinforcement Learning

To model environments, we consider Partially Observable Markov Decision Processes (POMDPs) [2], which are defined by a tuple $\langle S, A, T, R, \Omega, O, \gamma \rangle$. $S$, $A$, and $\Omega$ respectively represent the set of states, actions, and observations. $T(s_{t+1}|a_t, s_t)$ is the conditional transition distribution over the next state $s_{t+1} \in S$ given previous action $a_t \in A$ and state $s_t \in S$. Each transition produces an observation $o_{t+1} \in \Omega$ sampled from $O(o_{t+1}|s_{t+1}, a_t)$, and a reward $r_{t+1} \in \mathbb{R}$ sampled from $R(r_{t+1}|s_{t+1}, a_t, s_t)$.

An agent interacting with the POMDP observes $o_t$ and maintains a belief state $b_t$ based on the history of observations and actions. The agent then selects an action $a_t$ according to its policy. Reinforcement learning aims to find the optimal policy — a conditional distribution $\pi(a_t|h_t)$ that maximizes the expected discounted return $\mathbb{E}\left[\sum_{t=0}^{\infty} \gamma^t r_t\right]$, where $r_t$ is the reward received at time step $t$, $\gamma \in [0, 1)$ is the discount factor, and $h_t$ represents the history of observations and actions up to time $t$. In offline reinforcement learning, the agent cannot interact directly with the POMDP, but is given a dataset $\mathcal{D}$ of previously recorded interactions $(o_t, a_t, r_{t+1}, o_{t+1})$. There are no restrictions on how $\mathcal{D}$ is collected, so it may include data from a mix of policies with varying levels of expertise and state space coverage. Generalizing to states that differ from those in $\mathcal{D}$ is the main challenge of offline RL.

### 2.2 World Models

A world model [27] is a parameterized model of an environment that can be used to train reinforcement learning agents and autoregressively generate synthetic data. World models are able to generate imaginary trajectories by sampling from the following joint distribution:

$$p(s_{t+1:T}, r_{t+1:T}, a_{t:T-1}|s_t) = \prod_{i=t}^{T-1} \pi(a_i|s_i)T(s_{i+1}|s_i, a_i)R(r_{i+1}|s_i, a_i). \tag{1}$$

Creating a world model for a POMDP requires modeling the transition and observation distributions $T$ and $O$, and reward distribution $R$. Given a dataset $\mathcal{D}$ of interactions, we optimize the model parameters to minimize an expected loss

$$\min_{\hat{T}, \hat{O}, \hat{R}} \mathbb{E}_r\left[L_O(\hat{o}_t, o_t) + L_R(\hat{r}_t, r_t)\right], \tag{2}$$

where $\hat{o}_t$ and $\hat{r}_t$ are the world model predictions of $o_t$ and $r_t$ given $a_t, o_{t-1}$, and $L_O, L_R$ are appropriate loss functions. In practice, instead of predicting the next observation based on the prediction for the next state, we can predict it directly from the previous observation and action buffer, learning state transition dynamics implicitly via a hidden state (e.g., that of a recurrent neural network). Additionally, it is common to predict the residual in the observation $\Delta\hat{o}_{t+1} = \hat{o}_{t+1} - o_t$, which biases the world model towards temporal consistency in the autoregressive prediction.

### 2.3 Diffusion Models

Inspired by non-equilibrium thermodynamics, diffusion models [28] generate data by reversing a gradual noising process. The forward process corrupts a data sample $\mathbf{x}_0 \sim q(\mathbf{x}_0)$ into pure Gaussian noise $\mathbf{x}_T \sim \mathcal{N}(\mathbf{0}, \mathbf{I})$ through a Markov chain of Gaussian transitions:

$$q(\mathbf{x}_t \mid \mathbf{x}_{t-1}) = \mathcal{N}(\mathbf{x}_t; \sqrt{1 - \beta_t}\,\mathbf{x}_{t-1}, \beta_t\mathbf{I}), \tag{3}$$

where $\{\beta_t\}_{t=1}^T$ is a fixed variance schedule. The generative model learns the reverse process $p_\theta(\mathbf{x}_{t-1} \mid \mathbf{x}_t)$ by approximating the *score function*, i.e., the gradient of the log posterior $\nabla_{\mathbf{x}_t} \log q(\mathbf{x}_t \mid \mathbf{x}_0)$. In practice, a neural network $\epsilon_\theta(\mathbf{x}_t, t)$ is trained to predict the noise added during the forward process, and the training objective is typically:

$$\mathcal{L}_{\text{simple}} = \mathbb{E}_{\mathbf{x}_0, t, \epsilon}\left[\|\epsilon - \epsilon_\theta(\mathbf{x}_t, t)\|^2\right], \tag{4}$$

where $\epsilon \sim \mathcal{N}(\mathbf{0}, \mathbf{I})$ and $\mathbf{x}_t = \sqrt{\bar{\alpha}_t}\,\mathbf{x}_0 + \sqrt{1 - \bar{\alpha}_t}\,\epsilon$ with $\bar{\alpha}_t = \prod_{s=1}^t (1 - \beta_s)$.

This score matching framework, introduced by [29], enables estimation of score models directly from empirical data without requiring explicit knowledge of the true underlying score function. While standard diffusion models typically function as unconditional generators of a data distribution $p_{\text{data}}(x)$, adapting them for sequential decision-making requires conditioning on past information to predict future states.

To employ diffusion models as world models, we extend $\epsilon_\theta$ to model the environment's dynamics $p(x_{t+1}|x_{\leq t}, a_{\leq t})$, where $x_{t+1}$ is the next observation, while $x_{\leq t}$ and $a_{\leq t}$ denote the history of past observations and actions. This adaptation is particularly relevant for POMDPs, where the true environmental state must be inferred from observation and action sequences. The modified score model $\epsilon_\theta$ is trained by extending the standard diffusion objective to:

$$L(\theta) = \mathbb{E} \left\| \epsilon_\theta(x_\tau^{t+1}, \tau, x_0^{\leq t}, a^{\leq t}) - \epsilon \right\|^2 \tag{5}$$

where $x_\tau^{t+1}$ is the noised version of the next observation at noise level $\tau$, and $\epsilon$ is the noise added during the forward process.

To generate a subsequent observation $x_{t+1}$, we iteratively solve the reverse-time Stochastic Differential Equation (SDE)—or an equivalent Ordinary Differential Equation (ODE)—guided by our history-conditioned score model. While various numerical solvers can be employed, there exists a practical trade-off: sample quality typically improves with increased Number of Function Evaluations (NFE), directly impacting the computational cost of the diffusion-based world model during inference.

## 2.4 Prioritized Level Replay

When training agents to solve procedurally generated environments, previous methods have commonly relied on continuously and uniformly sampling new levels during training. This can be sample-inefficient since many sampled levels do not contribute significantly to the agent's learning progress. As an alternative, Prioritized Level Replay (PLR) [24] has been proposed, where levels are replayed based on their potential to induce greater learning progress in the agent. In original PLR implementation, it maintains a buffer of previously encountered levels, including a score that represents an estimate of learning potential and a staleness factor that represents how long ago that level was sampled. There are many possible choices for computing the score — a simple yet effective one is given by the TD-error $\delta_t = r_t + V(s_{t+1}) - V(s_t)$. During training, we randomly select between sampling a randomly generated level with a selected episode length range randomly and sampling one from the buffer. When sampling from the buffer, a level $l_i$ is selected for replay with probability

$$l_i \sim (1 - \rho) \cdot P_S(l \mid \Lambda_{\text{seen}}, S) + \rho \cdot P_C(l \mid \Lambda_{\text{seen}}, C, c), \tag{6}$$

which is a mixture distribution of $P_S$ and $P_C$ with parameter $\rho$. They prioritize levels with high scores and high staleness, respectively. Here $\Lambda_{\text{seen}}$ is the replay buffer, $S$ is the level scores, $C$ is the last time each level was sampled, and $c$ is how many levels were sampled since the beginning of training. In our world model setting, we save both the initial state $s_0$ and the desired imagination horizon $h$ in the buffer. When the algorithm samples an entry from the buffer, it uses the same initial state $s_0$ to begin imagination and runs the world model forward for exactly $h$ steps. By interpreting world models as procedural generators of agent experience, we can directly apply PLR to prioritize the most valuable imagined experiences. This yields a novel method for model-based offline RL that efficiently leverages imagined trajectories.

## 3 Imagined Autocurricula

Our approach consists of three key components: (1) a diffusion world model trained from diverse offline data, (2) using this model to generate imagined random episode length rollouts for agent training, and (3) implementing an auto curriculum over these imagined environments.

### 3.1 Diffusion World Model Training

First, we collect a diverse visual offline dataset of different behaviors (expert, medium, and random) via behavior policy as $D = \{(o_i, a_i, r_i, o_i')\}_{i=1}^N$. Then we train the diffusion model as a conditional

generative model of environment dynamics, $p(x_{t+1}|x_{<t}, a_{<t})$, handling the POMDP setting where the true Markovian state is unknown and must be approximated from history. Our training objective follows Equation 5. Following DIAMOND [11], we utilize the Elucidating Diffusion Models (EDM) formulation [30] rather than the traditional DDPM [28] approach. This critical design choice enables our world model to remain stable over long time horizons with significantly fewer denoising steps. During training, we sample trajectory segments from our dataset and apply noise using a $\tau$-level perturbation kernel to create training pairs. To incorporate temporal context, we maintain a buffer containing $L = 4$ previous observations and actions as conditioning information. Our diffusion world model implements a standard 2D U-Net architecture [31] as the core denoising network. The temporal information flows through two pathways: past observations are concatenated channel-wise with the noisy observation being denoised, while action information is integrated via adaptive group normalization layers [32] within residual blocks [33]. We iteratively solve the reverse SDE using Euler's method with exactly 5 denoising steps. We also follow their approach of using full-image observations (64×64 3-channel for Procgen) rather than discrete latent space representations, as they demonstrated superior performance over other state-of-the-art world models that use discrete latent space. Given the similar image dimensions between Atari and our Procgen environment, we leveraged their extensively validated hyperparameter settings for our diffusion model. Their comprehensive ablation study showed these parameters achieve both notable image generation quality and computational efficiency, making them an ideal foundation for our work. This approach allows our diffusion model to capture fine visual details critical for reinforcement learning for Procgen agents, such as small rewards and enemies, while maintaining temporal consistency across long sequences.

## 3.2 Agent Training in Imagined Environments

With our diffusion world model trained, we implement auxiliary predictors for reward and termination signals using an ensemble of $E$ prediction heads to capture uncertainty during imagined rollouts. While the base architecture follows Alonso et al. [11], employing a separate model $R_\psi$ with standard CNN [34] and LSTM [35] layers to handle partial observability, we extend their approach through our ensemble of prediction heads, which provides crucial uncertainty estimates during long-horizon planning in our POMDP setting. This model is trained on the same offline dataset used for the world model, optimizing for accurate reward and episode termination predictions, with the agent using the mean of the ensemble predictions during training. Building upon these diffusion and auxiliary models, the RL agent follows an actor-critic architecture parameterized by a shared CNN-LSTM backbone with separate policy and value function heads. The agent is trained using Advantage Actor-Critic (A2C) [36] with $\lambda$-returns for advantage estimation as in [37]. The policy head $\pi_\phi$ is optimized to maximize expected returns while the value network $V_\phi$ minimizes the temporal difference error between predicted values and the computed $\lambda$-returns.

During training, we generate imagined trajectories by first sampling an initial state $s_0$ from our diverse offline dataset, then autoregressively producing sequences of states, rewards, and termination signals by rolling out our policy through the diffusion denoiser model and ensemble auxiliary predictors. Unlike the fixed horizon approach in [11], we randomly sample the imagined episode length between minimum and maximum horizon bounds to incorporate greater diversity in training experiences and prevent the agent from exploiting fixed-horizon dynamics. Finally, we compute policy gradients and value updates based on these imagined experiences, enabling effective policy learning without additional environment interaction.

## 3.3 Autocurriculum

Our IMAC implementation follows the PLR framework [24], maintaining a replay buffer of previously encountered initial states $s_0$ with associated priority scores. For each training iteration, with probability 0.5, we train the agent using the current policy and world model on initial states sampled from this prioritized buffer. With the remaining 0.5 probability, we select a random initial state and a random imagined episode length, $T$, to explore the environment space and update the buffer. The priority score for each initial state $s_0$ is computed using the agent's temporal difference errors, $\delta_k$, which serve as a proxy for learning potential. The specific formula for the score is:

$$\text{score}(s_0) = \frac{1}{T} \sum_{t=0}^{T} \sum_{k=t}^{T} (\gamma\lambda)^{k-t} \max(0, \delta_k). \tag{7}$$

Here, $T$ is the length of the imagined episode (which is randomly chosen during the exploration phase), $\delta_k$ is the temporal difference error at timestep $k$, $\gamma$ is the discount factor, and $\lambda$ is the trace decay parameter. The inner sum, $\sum_{k=t}^{T}(\gamma\lambda)^{k-t}\max(0,\delta_k)$, represents a truncated, discounted, and $\lambda$-weighted sum of future *positive* TD errors from timestep $t$ to $T$. We specifically focus on positive TD errors through the $\max(0,\delta_k)$ operation because they indicate instances where the agent's value estimate was too low, suggesting unexpectedly good outcomes that are particularly valuable for learning. The outer sum and division by $T$ average this sum over all timesteps $t$ in the episode. Based on this prioritization mechanism, we believe that our approach creates three key advantages for generalization: First, by focusing on the scenarios with the highest learning potential, the agent spends more computation on informative experiences rather than those where little can be learned. Second, the random horizon setting creates diversity in training experiences, preventing the agent from exploiting fixed-length dynamics. Finally, the prioritization mechanism naturally discovers an emergent curriculum that gradually increases in difficulty as the agent improves. The emergent curriculum effect is particularly important for procedurally generated environments, where the difficulty distribution can be complex and unknown a priori. Our IMAC approach discovers this difficult landscape dynamically during training, focusing on the boundary of the agent's current capabilities—what PLR authors call "threshold levels" that provide the highest learning signal. As the agent improves, this boundary shifts to more challenging scenarios, creating a curriculum that scales with the agent's abilities without requiring manual design or environment-specific knowledge. Importantly, unlike previous autocurriculum learning approaches that require direct control of the environment generation process, our method works entirely within the imagined random-length rollouts from our world model, which we call our imagined autocurricula, making it applicable to any environment where a world model can be trained, including those where the procedural generation process is a black box. Details of model architecture, algorithm, training hyperparameters used, and dataset details for this work are given in Appendices A and B, C, and D.

## 4 Experiments

### 4.1 Procgen Benchmark

For comprehensive evaluation of our approach, we use a challenging subset of the Procgen Benchmark [38], a collection of procedurally generated environments designed specifically to test generalization in reinforcement learning. Unlike the Atari benchmark used in previous world model studies [11, 16, 15, 17, 39], Procgen environments are procedurally generated and thus inherently test an agent's ability to generalize to unseen levels with the same underlying mechanics.

We created a mixed offline dataset capturing diverse interaction patterns to represent the type of diverse data we may see in Internet videos. Similar to "mixed" datasets in existing benchmarks [40, 22] it consists of three complementary components: Expert trajectories ($D_{\text{expert}}$) from fully trained PPO agents, Medium-quality trajectories ($D_{\text{medium}}$) from partially trained agents reaching approximately 50% of expert performance, and Random exploration trajectories ($D_{\text{random}}$) to ensure broad state space coverage. This mixture forms a dataset $D = D_{\text{random}} \cup D_{\text{expert}} \cup D_{\text{medium}}$ totaling 1 million transitions per game across 200 training levels. We focus on seven environments from Procgen: CoinRun, Ninja, Jumper, Maze, and CaveFlyer, selected based on their poor performance in previous offline RL benchmarks [21] and their sparse reward structure. For each environment, we use 200 levels for training our world model while evaluating generalization on unseen test levels (201-$\infty$). After data collection, we trained the world model and reward/termination predictors (requiring approximately 10 hours on an NVIDIA RTX 4090), then froze these components for generating imagined rollouts during agent training. Each agent seed's training required approximately 4 days on a single RTX 4090, with experiments conducted across 3 random seeds for 5 million total steps each, resulting in 180 GPU days of computation.

We compare our approach with several established offline RL methods including Behavior Cloning (BC), Conservative Q-Learning (CQL) [41], Implicit Q-Learning (IQL) [42], Batch-Constrained Q-learning (BCQ) [43], Behavior-Constrained Transformers (BCT), and Decision Transformers (DT) [44]. We also conducted an ablation study comparing our approach against world model baselines, evaluating the impact of both fixed (15 steps) and variable (between 5 and 22 steps) imagined episode lengths.

## 4.2 Procgen Benchmark Results

Table 1: **Generalization results**: All values represent the mean return over three random seeds ± standard deviation when transferring agents to held out levels on the Procgen benchmark.

| Method | CoinRun | Ninja | Jumper | Maze | CaveFlyer | Heist | Miner |
|---|---|---|---|---|---|---|---|
| BC | $5.31 \pm 0.31$ | $4.22 \pm 0.21$ | $3.92 \pm 0.42$ | $4.03 \pm 0.29$ | $3.12 \pm 0.17$ | $2.1 \pm 0.22$ | $4.68 \pm 0.11$ |
| CQL | $5.12 \pm 0.22$ | $3.29 \pm 0.32$ | $2.24 \pm 0.31$ | $1.82 \pm 0.12$ | $1.89 \pm 0.27$ | $0.3 \pm 0.09$ | $0.32 \pm 0.09$ |
| IQL | $4.96 \pm 0.20$ | $3.11 \pm 0.31$ | $2.35 \pm 0.38$ | $1.98 \pm 0.17$ | $1.45 \pm 0.47$ | $0.27 \pm 0.04$ | $0.5 \pm 0.08$ |
| BCQ | $4.33 \pm 0.14$ | $3.14 \pm 0.26$ | $2.29 \pm 0.18$ | $1.83 \pm 0.14$ | $2.21 \pm 0.61$ | $0.44 \pm 0.12$ | $0.42 \pm 0.13$ |
| BCT | $5.11 \pm 0.10$ | $4.33 \pm 0.28$ | $3.72 \pm 0.37$ | $3.98 \pm 0.19$ | $2.76 \pm 0.22$ | $1.5 \pm 0.23$ | $1.08 \pm 0.10$ |
| DT | $5.03 \pm 0.18$ | $4.03 \pm 0.31$ | $3.84 \pm 0.24$ | $4.32 \pm 0.11$ | $3.04 \pm 0.32$ | $1.82 \pm 0.09$ | $1.11 \pm 0.09$ |
| **iMac** | $\mathbf{6.20} \pm 0.14$ | $\mathbf{5.15} \pm 0.12$ | $\mathbf{5.78} \pm 0.21$ | $\mathbf{5.41} \pm 0.19$ | $\mathbf{3.87} \pm 0.22$ | $\mathbf{2.91} \pm 0.15$ | $\mathbf{5.61} \pm 0.13$ |

Table 2: **World model ablation**: Performance evaluation on the Procgen benchmark all values represent mean return over three random seeds ± standard deviation

| Method | CoinRun | Ninja | Jumper | Maze | CaveFlyer | Heist | Miner |
|---|---|---|---|---|---|---|---|
| WM Fixed | $4.76 \pm 0.19$ | $3.78 \pm 0.21$ | $3.71 \pm 0.19$ | $3.92 \pm 0.32$ | $2.94 \pm 0.13$ | $2.05 \pm 0.14$ | $3.18 \pm 0.10$ |
| WM Random | $5.36 \pm 0.15$ | $4.33 \pm 0.18$ | $4.11 \pm 0.11$ | $4.62 \pm 0.15$ | $3.02 \pm 0.19$ | $2.19 \pm 0.18$ | $4.55 \pm 0.15$ |
| **iMac** | $\mathbf{6.20} \pm 0.14$ | $\mathbf{5.15} \pm 0.12$ | $\mathbf{5.78} \pm 0.21$ | $\mathbf{5.41} \pm 0.19$ | $\mathbf{3.87} \pm 0.22$ | $\mathbf{2.91} \pm 0.15$ | $\mathbf{5.61} \pm 0.13$ |

The results in Table 1 demonstrate that IMAC method consistently outperforms state-of-the-art offline RL algorithms across all evaluated Procgen environments. Our approach achieves improvements of 17% on CoinRun, 19% on Ninja, 48% on Jumper, 35% on Maze, 24% on CaveFlyer, 38% on Heist, and 19% on Miner compared to the best-performing model-free baseline for each environment.

Table 2 further demonstrates IMAC's effectiveness through ablation studies comparing different horizon-setting strategies. While both fixed and random episode length world model approaches show strong performance, our IMAC method incorporating prioritized initial state selection consistently delivers improved results—achieving up to 56% improvement over the fixed horizon baseline on Jumper and 38% on Maze. These results provide compelling evidence that procedural generalization in world models benefits significantly from our autocurriculum approach, which automatically discovers and focuses training on the most informative scenarios. The substantial performance advantage across diverse environments demonstrates that IMAC effectively addresses the core challenge of generalization from limited offline datasets in procedurally generated environments. We provide all the hyperparameters used in model-free baseline algorithms in Appendix F.

To provide a comprehensive statistical analysis of our approach's effectiveness, Figure 2 presents the aggregated mean performance improvements across all evaluated environments, offering quantitative evidence of the consistent generalization benefits achieved by our method compared to the baselines presented in both tables.

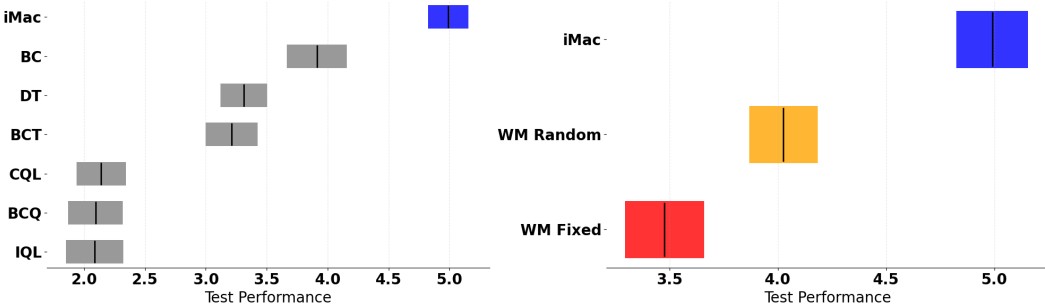

Figure 2: Mean test performance across seven Procgen environments. Left: Comparison of model-free baselines against our proposed IMAC method. Right: World model ablation study comparing WM variants (Fixed and Random) against the full iMac approach

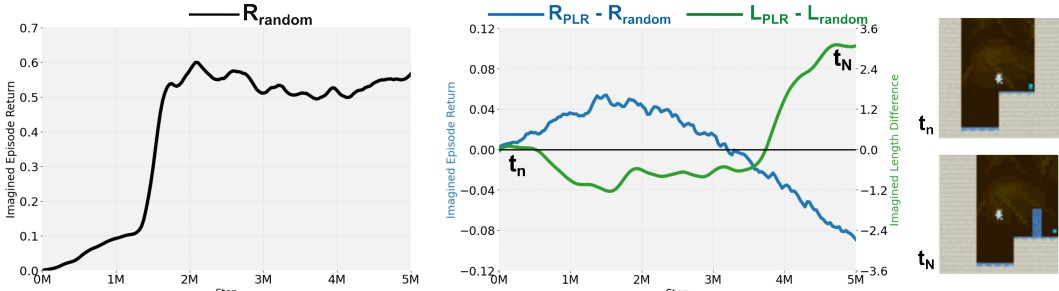

Figure 3: Evolution of performance and episode length during training with our Imagined Autocurricula approach. **Left:** Mean episodic return of the random baseline ($R_{Random}$) showing steady improvement over training steps. **Right:** The difference between PLR and Random approaches in terms of both return ($R_{PLR} - R_{Random}$) and imagined episode length ($L_{PLR} - L_{Random}$).

Our results provide compelling evidence for the effectiveness of the emergent curriculum created by our Imagined Autocurricula approach. As shown in Figure 3, initially PLR selects similar episode lengths as random sampling while achieving higher returns for CoinRun. However, after approximately 4M steps, a significant shift occurs—PLR begins prioritizing substantially longer episodes while temporarily accepting lower comparative returns. This demonstrates the emergent curriculum effect: our prioritization mechanism automatically discovers and focuses on more challenging, temporally-extended scenarios aps the agent's capabilities improve.

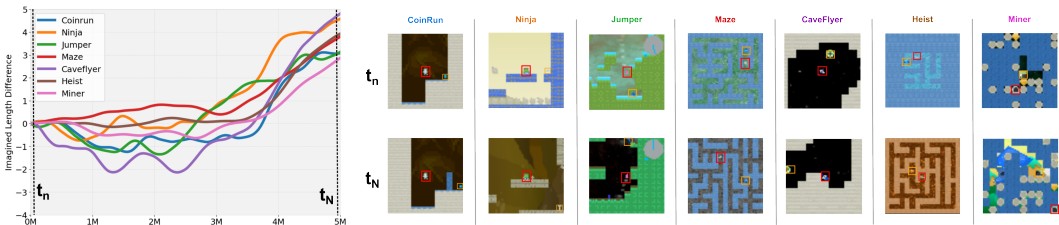

Figure 4: Left: Difference in imagined episode length between PLR and Random baselines across seven Procgen environments. Right: Example frames from early $t_n$ and late $t_N$ training stages, with red boxes representing agents and orange boxes representing rewards.

The visual examples in Figure 4 further illustrate this curriculum effect across all environments. Comparing initial states $t_0$ with later states $t_N$, we observe a clear progression in complexity—from simple, direct paths to complex obstacle arrangements for CoinRun, and from straightforward corridors to intricate maze structures requiring extended planning in Maze. The image pairs show representative states from early and late training stages, illustrating the progression from simpler to more complex scenarios. This progressive increase in horizon differences correlates directly with our method's performance on test environments, supporting our hypothesis that learning temporally extended behaviors is crucial for generalization in procedurally generated environments. What's particularly notable is that this curriculum emerges naturally from our prioritization mechanism without any explicit difficulty programming or environment-specific knowledge. Our method discovers this effective learning progression while training exclusively on imagined rollouts from a fixed world model trained on offline data, validating our core claim that prioritizing states based on their learning potential substantially improves generalization capabilities.

## 5 Related Work

**Generalization in Reinforcement Learning** has been extensively studied, though primarily in online settings where agents can actively collect new data during training [45, 46, 47, 38, 48, 49, 50, 51, 52, 53, 54, 55, 56, 57, 58, 59]. Benchmarks for evaluating generalization capabilities have evolved from environment suites like Procgen [38] and NetHack [41] for online settings, to more recent offline alternatives such as V-D4RL and OFFLINE PROCGEN [22, 21]. These offline benchmarks

present unique challenges, particularly in visual domains where balancing dataset scale and accessibility becomes crucial—large-scale datasets like Atari, StarCraft, and MineRL contain millions of samples but require substantial computational resources [60, 61, 62], while more accessible alternatives offer 100,000 to 1 million samples per environment. Despite this progress, generalization in offline RL remains relatively unexplored compared to its online counterpart, especially when dealing with procedurally generated environments where model-free methods have shown significant limitations. Our work addresses this gap by leveraging world models to improve generalization across diverse procedurally generated environments without requiring additional environment interaction beyond the initial offline dataset.

**World Models** have evolved significantly since the first introduction of the concept of reinforcement learning within an imagined neural network environment [27]. Early applications to Atari games through SimPLe [39] established the Atari 100k benchmark, highlighting the sample efficiency benefits of model-based approaches. The field advanced substantially with DreamerV2 [17], which pioneered learning directly from the latent space of a recurrent state space model (RSSM), while DreamerV3 [12] achieved human-level performance across diverse domains using consistent hyperparameters. Recent architectural innovations include transformer-based approaches such as TWM [16] and STORM [32], which adapt DreamerV2's and DreamerV3's RSSM frameworks respectively, employing different tokenization strategies. IRIS [15] represents an alternative approach, constructing a language of image tokens through a discrete autoencoder and composing these tokens temporally using an autoregressive transformer. Most recently, DIAMOND [11] introduced diffusion models as an alternative to discrete latent representations for world modeling, demonstrating that improved visual fidelity can lead to better agent performance, particularly for tasks where fine visual details are critical. Recent advances in video generation, including 3D DiT architectures like CogVideoX [63], Cosmos [9], and realistic world generators like Genie 3 [8] offer promising alternatives for world modeling. While we employ a 2D UNet for computational efficiency, our curriculum learning framework is architecture- agnostic and could leverage these more advanced architectures as they become computationally more feasible. There has also been work using world models to generalize to unseen tasks [64, 65], however these were focused on a domain randomization setting (e.g. our "random" baseline) rather than an automated curriculum. By contrast, our work builds upon these foundations and leverages PLR to generate an emergent curriculum, which subsequently improves generalization to unseen tasks.

**Curriculum Learning in RL** applies the principle of gradually increasing task difficulty to reinforcement learning, improving both training efficiency and generalization capabilities [66, 67]. Determining learning priorities is central to effective curriculum design, with various measures including reward signals [68], learning progress [69], TD-errors [70], and state novelty [71]. Recent advances in automated curriculum learning [72] have reduced reliance on manual design, with methods like Prioritized Level Replay (PLR) [24] demonstrating that targeting environments with the highest learning potential creates an emergent curriculum that progressively challenges the agent at the frontier of its capabilities. While we employ PLR in our implementation, our framework is compatible with other UED algorithms including ACCEL [73], which uses evolution strategies for curriculum design; ADD [74], which leverages diffusion models for adversarial environment generation. These methods could potentially be integrated into our framework, highlighting the modularity of our approach. Our work applies this curriculum-based approach to world models, creating an imagined autocurriculum that operates entirely within generated rollouts from a fixed world model trained on offline data, enabling effective generalization without requiring explicit environment control or online interaction.

## 6   Limitations

While our approach demonstrates promising results, we acknowledge certain constraints inherent to offline world modeling. The quality of generated scenarios naturally depends on training data diversity, though our mixed-expertise dataset mitigates this concern in practical settings. Implementation of IMAC requires substantial computational resources, and its effectiveness varies across environment types. Future work could address these limitations through efficiency optimizations, domain-adaptive world modeling techniques, automated parameter tuning, and improved uncertainty quantification. Additional promising directions include innovative data augmentation strategies and extensions to continuous control domains with higher-dimensional state spaces—all building upon our demonstrated foundation. We provide computational cost analysis for our method in Appendix G.

# 7 Conclusion

We introduced IMAC, a novel approach combining diffusion-based world models with automatic curriculum learning to train general agents. Our method leverages offline data to generate diverse imagined environments and employs Prioritized Level Replay to create an emergent curriculum that adapts to the agent's capabilities. Experimental results across seven Procgen environments demonstrate that IMAC consistently outperforms state-of-the-art offline RL algorithms and world model baselines, achieving improvements of up to 48% compared to model-free approaches and 56% over fixed-horizon world models. The emergent curriculum proves particularly effective for environments requiring extended planning horizons and complex navigation. Our work demonstrates that strong transfer performance on held-out environments is possible using only imagined trajectories from a world model trained on a limited dataset. This finding opens a promising path toward utilizing larger-scale foundation world models for developing generally capable agents that can handle novel task variations.

Future work should address the current limitations of our approach by exploring techniques to reduce computational requirements, investigating novel curriculum learning methodologies with world model architectures with stronger uncertainty quantification, and developing methods to automatically adapt prioritization parameters based on environment characteristics. Additionally, extending IMAC to continuous control domains and environments with higher complexity or extremely sparse rewards represents an important direction for expanding the applicability of our method.

# 8 Acknowledgements

This work was supported by the Engineering and Physical Sciences Research Council (EPSRC) EP/S021566/1.

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

# A  Model Architectures

**Diffusion World Model**  The diffusion model $D_\theta$ employs a standard 2D U-Net architecture [31] for next-frame prediction. The model conditions on a history window of 4 previous frames and their corresponding actions, along with the diffusion timestep $\tau$. We implement observation conditioning through frame stacking along the channel dimension, while action and diffusion time conditioning are incorporated via adaptive group normalization layers [32] within the U-Net's residual blocks.

**Reward and Termination Predictor**  The reward/termination model $R_\psi$ utilizes a shared backbone architecture with separate prediction heads for reward and termination signals. Crucially, we employ an ensemble of 10 prediction heads to enable uncertainty quantification during imagination rollouts. The model processes sequences of frames and actions through convolutional residual blocks [33] followed by an LSTM cell [36, 35]. During inference, the ensemble predictions are aggregated to compute both mean estimates and uncertainty measures, providing robustness against prediction errors in long-horizon imagined trajectories. Prior to imagination, we perform burn-in [75] using the conditioning frames and actions to properly initialize the LSTM's hidden and cell states.

**Actor-Critic Network**  The policy $\pi_\phi$ and value network $V_\phi$ share a common feature extraction backbone with separate output layers. We refer to this combined architecture as the actor-critic network $(\pi, V)_\phi$, though technically $V$ represents a state-value function rather than an action-value critic. The network processes individual frames through a convolutional encoder consisting of four residual blocks with $2\times2$ max-pooling (stride 2) between blocks. Each residual block contains a main path with group normalization [76], SiLU activation [77], and $3\times3$ convolution (stride 1, padding 1). The convolutional features are then processed by an LSTM cell to maintain temporal context. Similar to the reward model, we initialize the LSTM states through burn-in on conditioning frames before beginning the imagination procedure.

Please refer to Table 3 below for hyperparameter values.

Table 3: Architecture Parameters for iMAC.

| Hyperparameter | Value |
| --- | --- |
| **Diffusion Model ($D_\theta$)** | |
| Observation conditioning mechanism | Frame stacking |
| Action conditioning mechanism | Adaptive Group Normalization |
| Diffusion time conditioning mechanism | Adaptive Group Normalization |
| Residual blocks layers | [2, 2, 2, 2] |
| Residual blocks channels | [64, 64, 64, 64] |
| Residual blocks conditioning dimension | 256 |
| **Reward/Termination Model ($R_\psi$)** | |
| Action conditioning mechanism | Adaptive Group Normalization |
| Residual blocks layers | [2, 2, 2, 2] |
| Residual blocks channels | [32, 32, 32, 32] |
| Residual blocks conditioning dimension | 128 |
| LSTM dimension | 512 |
| **Actor-Critic Model ($\pi_\phi$ and $V_\phi$)** | |
| Residual blocks layers | [1, 1, 1, 1] |
| Residual blocks channels | [32, 32, 64, 64] |
| LSTM dimension | 512 |

# B    World Model Training Parameters

Table 4: Hyperparameters for ɪMAC.

| Hyperparameter | Value |
|---|---|
| **Training loop** | |
| Number of epochs | 1000 |
| Training steps per epoch | 100 |
| Batch size | 32 |
| Environment steps per epoch | 100 |
| Epsilon (greedy) for collection | 0.01 |
| **RL hyperparameters** | |
| Imagination horizon ($H$) | Fixed:15, Random:[5-22] |
| Discount factor ($\gamma$) | 0.985 |
| Entropy weight ($\eta$) | 0.001 |
| $\lambda$-returns coefficient ($\lambda$) | 0.95 |
| **Sequence construction during training** | |
| For $D_\theta$, number of conditioning observations and actions ($L$) | 4 |
| For $R_\psi$, burn-in length ($B_R$), set to $L$ in practice | 4 |
| For $R_\psi$, training sequence length ($B_R + H$) | Fixed:19, Random[9-26] |
| For $\pi_\phi$ and $V_\phi$, burn-in length ($B_{\pi,V}$), set to $L$ in practice | 4 |
| **Optimization** | |
| Optimizer | AdamW |
| Learning rate | 4e-5 |
| Epsilon | 1e-8 |
| Weight decay ($D_\theta$) | 1e-2 |
| Weight decay ($R_\psi$) | 1e-2 |
| Weight decay ($\pi_\phi$ and $V_\phi$) | 5e-5 |
| **Diffusion Sampling** | |
| Method | Euler |
| Number of steps | 5 |
| **Environment** | |
| Image observation dimensions | 64×64×3 |
| Action space | Discrete (up to 18 actions) |
| Frameskip | 4 |
| Max noop | 30 |
| Termination on life loss | True |
| Reward clipping | $\{0, 1\}$ |
| **PLR** | |
| Stalaness Factor | 0.1 |
| Buffer Size | 2500 |

# C Baseline World Model and Actor-Critic Training Algorithm

---

**Algorithm 1:** iMac (Sequential Offline Training)

---

**Procedure** *training_loop(D)*
    **//** $D$ **is a fixed offline dataset**;
    **Phase 1: Train Diffusion Model**;
    **for** *epochs_diffusion **do*** **do**
        **for** *steps_diffusion_model **do*** **do**
            update_diffusion_model($D$);
        **end**
    **end**
    **// Freeze Diffusion Model** $D_\theta$;
    **Phase 2: Train Reward/Termination Model**;
    **for** *epochs_reward **do*** **do**
        **for** *steps_reward_end_model **do*** **do**
            update_reward_end_model($D$);
        **end**
    **end**
    **// Freeze Reward/Termination Model** $R_\psi$;
    **Phase 3: Train Actor-Critic with Frozen Models**;
    **for** *epochs_actor_critic **do*** **do**
        **for** *steps_actor_critic* **do**
            update_actor_critic($D$);
            **// Uses frozen** $D_\theta$ **and** $R_\psi$
        **end**
    **end**

**Procedure** *update_diffusion_model(D)*
    Sample sequence $(x^0_{t-L+1}, a_{t-L+1}, \ldots, x^0_t, a_t, x^0_{t+1}) \sim D$;
    Sample $\log(\sigma) \sim \mathcal{N}(F_{mean}, F^2_{std})$ **//** log-normal sigma distribution from EDM;
    Define $\tau := \sigma$ **//** default identity schedule from EDM;
    Sample $x^1_{t+1} \sim \mathcal{N}(x^0_{t+1}, \sigma^2 I)$ **//** Add independent Gaussian noise;
    Compute $x^0_{t+1} = D_\theta(x^1_{t+1}, \tau, x^0_{t-L+1}, a_{t-L+1}, \ldots, x^0_t, a_t)$;
    Compute reconstruction loss $\mathcal{L}(\theta) = ||x^0_{t+1} - x^1_{t+1}||^2$;
    Update $D_\theta$;

**Procedure** *update_reward_end_model(D)*
    Sample indices $I := \{t, \ldots, t + L + H - 1\}$ **//** burn-in + imagination horizon;
    Sample sequence $(x^0_i, a_i, r_i, d_i)_{i \in I} \sim D$;
    Initialize $h = c = 0$ **//** LSTM hidden and cell states;
    **for** $i \in I$ **do** **do**
        Compute $\hat{r}_i, \hat{d}_i, h, c = R_\psi(x_i, a_i, h, c)$;
    **end**
    Compute $\mathcal{L}(\psi) = \sum_{i \in I} \text{CE}(r_i, \text{sign}(r_i)) + \text{CE}(d_i, \hat{d}_i)$ **//** CE: cross-entropy loss;
    Update $R_\psi$;

**Procedure** *update_actor_critic(D)*
    Sample initial buffer $(x^0_{t-L+1}, a_{t-L+1}, \ldots, x^0_t) \sim D$;
    Burn-in buffer with $R_\psi, \pi_\phi$ and $V_\phi$ to initialize LSTM states;
    **for** $i = t$ *to* $t + H - 1$ **do** **do**
        Sample $a_i \sim \pi_\phi(a_i | x^0_i)$;
        Sample reward $r_i$ and termination $d_i$ with frozen $R_\psi$;
        Sample next observation $x^0_{i+1}$ by simulating reverse diffusion process with frozen $D_\theta$;
    **end**
    Compute $V_\phi(x_i)$ for $i = t, \ldots, t + H$;
    Compute RL losses $\mathcal{L}_V(\phi)$ and $\mathcal{L}_\pi(\phi)$;
    Update $\pi_\phi$ and $V_\phi$;

**Algorithm 2:** iMac: Actor-Critic Training with PLR (Prioritized Level Replay)

**Procedure** *train_actor_critic_PLR(D, B)*

    **//** $D$ **is the offline dataset,** $B$ **is the prioritized buffer**;

    **for** *epochs_actor_critic* ***do*** **do**

        **// Update buffer with probability 0.5**;

        **if** *random()* $< 0.5$ ***then*** **then**

            | update_PLR_buffer($D$, $B$);

        **end**

        **for** *steps_actor_critic* ***do*** **do**

            **// Train actor-critic using only buffer data**;

            update_actor_critic_from_buffer($B$);

        **end**

    **end**

**Procedure** *update_PLR_buffer(D, B)*

    **// Sample horizon length randomly**;

    Sample initial sequence $(x_{t-L+1}^0, a_{t-L+1}, \ldots, x_t^0) \sim D$;

    Burn-in buffer with $R_\psi, \pi_\phi$ and $V_\phi$ to initialize LSTM states;

    **// Imagine trajectory using frozen models**;

    Initialize trajectory $\tau = \{\}$;

    **for** $i = t$ *to* $t + H - 1$ ***do*** **do**

        Sample $a_i \sim \pi_\phi(a_i | x_i^0)$;

        Sample reward $r_i$ and termination $d_i$ with frozen $R_\psi$;

        Sample next observation $x_{i+1}^0$ by simulating reverse diffusion process with frozen $D_\theta$;

        Compute TD-error $\delta_i = r_i + \gamma V_\phi(x_{i+1}^0) - V_\phi(x_i^0)$;

        $\tau \leftarrow \tau \cup \{(x_i^0, a_i, r_i, d_i, x_{i+1}^0, \delta_i)\}$;

        **if** $d_i = 1$ ***then*** **then**

            | **break**;

        **end**

    **end**

    **// Calculate PLR score as positive mean TD-error**;

    Compute PLR score $s_\tau = \text{mean}(\max(\delta_i, 0) \text{ for } \delta_i \text{ in } \tau)$;

    **// Update buffer if score is higher than any existing entry**;

    **if** $s_\tau > \min(s_b \text{ for } b \in B)$ ***then*** **then**

        | Replace lowest-scoring entry in $B$ with $(\tau, s_\tau)$;

    **end**

**Procedure** *update_actor_critic_from_buffer(B)*

    **// Sample trajectory from buffer based on priorities**;

    Sample trajectory $\tau \sim B$ according to PLR scores;

    **// Extract states, actions, rewards, etc.**;

    Extract $(x_i^0, a_i, r_i, d_i, x_{i+1}^0)$ from $\tau$;

    **// Compute values for all states in trajectory**;

    Compute $V_\phi(x_i^0)$ for all states in $\tau$;

    **// Compute advantage estimates and returns**;

    Compute advantages $A_i$ and returns $G_i$ for all transitions;

    **// Update actor and critic**;

    Compute RL losses $\mathcal{L}_V(\phi) = \sum_i (V_\phi(x_i^0) - G_i)^2$;

    Compute $\mathcal{L}_\pi(\phi) = -\sum_i \log \pi_\phi(a_i | x_i^0) \cdot A_i$;

    Update $\pi_\phi$ and $V_\phi$;

# D    Dataset Collection Details

We constructed our offline dataset by collecting trajectories from 200 procedurally generated levels for each of the five Procgen environments (CoinRun, Ninja, Jumper, Maze, and CaveFlyer), totaling 1 million environment steps per game. We explicitly constrained data collection to levels 0-199 for each environment, ensuring complete isolation from the test levels (200+) used for evaluation, thereby preventing any potential data leakage and maintaining a strict train-test split that properly assesses generalization to truly unseen procedurally generated scenarios.Following and extending the benchmark protocol from the latest work [21], we created a mixed-quality dataset with equal proportions (1/3 each) of expert, medium, and random trajectories. Expert trajectories were collected from fully-trained PPO agents [36], medium trajectories from agents achieving 50% of expert performance, and random trajectories from uniformly random action selection. Expert trajectories were collected from PPO agents trained until convergence (when learning plateaued) rather than for a fixed number of steps, ensuring optimal performance for each game's unique characteristics.This mixed composition introduces additional challenges compared to standard offline RL benchmark [21] by reducing the proportion of successful episodes. The dataset is stored in standard (s, a, r, s') format where observations s are 64×64×3 RGB images, actions a vary by environment-specific action spaces, and rewards are sparse—appearing only at episode termination in all five selected environments. This extreme sparsity in reward signals presents a particularly challenging scenario for world model learning, as the model must learn long-horizon dynamics without intermediate reward guidance. The hyperparameters of behavior policy are provided at Appendix F.

# E    Analysis

## E.1    Emergent Curriculum Effect

Our results demonstrate a clear emergent curriculum effect through the Prioritized Level Replay mechanism. As shown in Figure 2, the PLR approach initially selects similar episode lengths to random sampling while achieving higher returns. However, after approximately 4M training steps, a significant shift occurs where PLR begins prioritizing substantially longer episodes (up to $2.4\times$ longer in some environments) while temporarily accepting lower comparative returns. This behavior indicates that the autocurriculum naturally discovers and focuses on more challenging, temporally-extended scenarios as the agent's capabilities improve, without requiring any manual curriculum design or environment-specific knowledge.

## E.2    Component Contributions

The ablation study in Table 2 reveals the critical importance of combining both PLR and random horizon sampling. While fixed-horizon world models show reasonable performance (e.g., 3.71 on Jumper), incorporating random horizons improves results by 11% (4.11), and adding PLR-based prioritization yields an additional 41% improvement (5.78). This suggests that the success of IMAC stems from the synergistic effect of (1) diverse training experiences through random horizons preventing overfitting to fixed-length dynamics, and (2) intelligent prioritization that focuses computational resources on the most informative imagined trajectories.

## E.3    Generalization Mechanisms

The superior generalization of IMAC can be attributed to its ability to generate and prioritize diverse imagined experiences beyond the offline dataset distribution. By training exclusively on imagined rollouts from a world model learned on 200 levels, our agents achieve strong performance on unseen test levels (201+), with improvements ranging from 17% to 48% over model-free baselines. This suggests that the combination of world models and autocurricula enables agents to explore a richer space of potential scenarios than what exists in the original offline data, effectively augmenting the training distribution in a principled manner.

# F Baseline Algorithm Hyperparameters

We conducted comprehensive hyperparameter optimization for all baseline algorithms using grid search following the previous benchmark work [21]. All model-free offline RL methods (BCQ, CQL, IQL) and behavior cloning (BC) shared identical encoder architectures, utilizing ResNet for visual feature extraction from the 64×64×3 Procgen observations. Experiments were executed on NVIDIA RTX 4090 GPUs with 24GB memory. For behavior cloning (BC), we explored batch sizes $\in \{64, 128, 256, 512\}$ and learning rates $\in \{5\times10^{-3}, 1\times10^{-4}, 5\times10^{-4}, 6\times10^{-5}\}$. The Q-learning based methods (BCQ, CQL, IQL) employed dual-network architectures with target networks, where we investigated both Polyak averaging and direct weight copying strategies. For direct copying, we tested update frequencies $\in \{1, 100, 1000\}$, while for Polyak averaging, we examined $\tau \in \{0.005, 0.5, 0.99\}$. Algorithm-specific hyperparameters included: BCQ's action selection threshold $\in \{0.3, 0.5, 0.7\}$; CQL's regularization coefficient $\alpha \in \{0.5, 1.0, 4.0, 8.0\}$; and IQL's temperature $\in \{3.0, 7.0, 10.0\}$ with expectile weights $\in \{0.7, 0.8, 0.9\}$. For sequence modeling approaches (DT, BCT), we additionally optimized context lengths $\in \{5, 10, 30, 50\}$. Decision Transformer utilized return-to-go multipliers $\in \{1, 5\}$, where the maximum return was determined from training data and scaled accordingly. Both DT and BCT maintained a dropout rate of 0.1 following [44]. Table 5 presents the final selected hyperparameters achieving optimal performance on our 1M-step mixed dataset.

Table 5: Final hyperparameter configurations for baseline algorithms on 1M mixed dataset

| Algorithm | Hyperparameter | Value |
|---|---|---|
| BC | Learning Rate | 0.0005 |
| | Batch Size | 256 |
| BCT | Learning Rate | 0.0005 |
| | Batch Size | 512 |
| | Context Length | 30 |
| | Eval Return Multiplier | 0 |
| DT | Learning Rate | 0.0005 |
| | Batch Size | 512 |
| | Context Length | 10 |
| | Eval Return Multiplier | 5 |
| BCQ | Learning Rate | 0.0005 |
| | Batch Size | 256 |
| | Target Model Weight Update | Direct copy |
| | $\tau$ | - |
| | Target Update Frequency | 1000 |
| | Threshold | 0.5 |
| CQL | Learning Rate | 0.0005 |
| | Batch Size | 256 |
| | Target Model Weight Update | Direct copy |
| | $\tau$ | - |
| | Target Update Frequency | 1000 |
| | Alpha | 4.0 |
| IQL | Learning Rate | 0.0005 |
| | Target Model Weight Update | Direct copy |
| | Batch Size | 512 |
| | $\tau$ | - |
| | Target Update Frequency | 100 |
| | Temperature | 3.0 |
| | Expectile | 0.8 |

# G  Training Time Profile

Table 6 presents a comprehensive breakdown of computational time requirements for training IMAC 1 game per seed.

Table 6: Detailed computational time analysis for IMAC training pipeline. Measurements were obtained using an NVIDIA RTX 4090 GPU with hyperparameters detailed in previous sections. These timing results serve as representative benchmarks, with actual runtimes varying based on hardware configuration, environment setup, and specific training conditions.

| Component | Time (hours) |
|---|---|
| **World Model Training (Offline)** | |
| Diffusion model | 6.8 |
| Reward/Termination model | 5.4 |
| **Agent Training** | |
| Total (1000 epochs) | 75.0 |
| Per epoch (avg) | 0.075 |
| **Total Training Time** | 87.2 |

