# OpenReview forum: "Imagined Autocurricula"
_NeurIPS.cc/2025/Conference — NeurIPS 2025 poster_

### Official Review · Reviewer_NSyY · 2025-06-24

**Clarity:** 3
**Significance:** 2
**Originality:** 2
**Rating:** 4
**Confidence:** 4

**Summary:**

This paper introduces Imagined Autocurricula (iMAC), a novel method for training reinforcement learning agents. The method leverages offline data to generate diverse imagined environments using a trained diffusion model and employs Prioritized Level Replay to create an emergent curriculum that adapts to the agent’s capabilities.

**Questions:**

The performance of IMAC is fundamentally tied to the quality of the generative world model, which in turn depends on the offline dataset used for its training. The paper uses a "mixed" dataset of expert, medium, and random trajectories. How does IMAC's performance degrade if the world model is trained on a less diverse or lower-quality dataset?

The authors should discuss the OMNI-EPIC paper, as it appears to make similar claims. A discussion comparing the two approaches would help to clarify the novel contributions of this paper.

**Ethical Concerns:**

["NO or VERY MINOR ethics concerns only"]

**Final Justification:**

The authors have adequately addressed my concerns.

**Limitations:**

Yes

**Quality:**

3

**Strengths And Weaknesses:**

## Strengths

The paper is clear and well-written. The introduction sets up the problem context and motivation effectively. The proposed method is explained logically, with a helpful architectural overview in Figure 1.

## Weaknesses

The quality of the diffusion model depends on PPO being able to solve the tasks during data collection. It is unclear if the final agent is able to solve qualitatively different tasks after being trained with iMac.

The paper claims to demonstrate "open-ended learning in learned world models". This is a powerful claim, as open-ended learning often implies the generation of tasks with ever-increasing and potentially unbounded complexity. Given that the world model is fixed after training on a finite dataset, the space of generated environments is necessarily constrained by what the model has learned.

A potential weakness is the complexity of the method with many moving pieces.

Another potential weakness is the very high computational cost—180 GPU days for the main experiments —which may be a barrier to reproducibility and adoption. However, the authors are transparent about this in the limitations section.

---

> ### Author Rebuttal · Authors · 2025-07-30
>
> We thank the reviewer for acknowledging our clear presentation and logical method explanation. We address the conceptual concerns below to clarify our claims and contributions.
>
> **_for waekness 1)_** **Qualitatively Different Tasks:** Our results actually demonstrate stronger task diversity than initially described. In Procgen environments, longer episode horizons represent qualitatively harder tasks, not just extended versions. Figure 4 shows our method discovers 2-4 step horizon increases during training, which translates to more complex tasks as demonstrated in the tn vs. tN visual examples. The progression from simple direct paths to complex multi-obstacle courses (CoinRun), basic corridors to intricate multi-room maze navigation (Maze) demonstrates **more obstacles, enemies, and spatial complexities the agent must overcome before reaching the goal**. Our PLR mechanism automatically discovers this difficulty progression without manual design, showing that agents learn to solve increasingly complex instances requiring fundamentally different strategies and capabilities. **We have also added two more additional Procgen diverse domains for camera ready version—Heist (puzzle-solving) and Miner (physics simulation)—achieving 19.8% and 38.5% improvements respectively.**
>
> **_for weakness 2)_** **Open-Ended Learning:** We appreciate the reviewer highlighting that we have not yet achieved open-ended learning in the absolute sense. Our claim should state that our work opens a path towards "open-ended learning in learned world models." which we believe is valid. Many would consider that UED algorithms fall within the general scope of “open-ended learning” and indeed PLR was used in the Adaptive Agents paper using XLand [1] which is undeniably an “open-ended learning” paper. As far as we are aware, we are the first to bring these ideas to learned world models. Of course, the world models we use are narrow in scope (modelling a distribution of procgen environments) **but we believe the ideas will transfer to larger-scale foundation world models, which are increasingly modelling vast and diverse, open-ended and even unlimited task spaces**. We believe our work could be a key stepping stone to using such models for open-ended learning with world model.
>
> **_for weakness 3)_** **Method Complexity:** While our approach integrates multiple components, each serves a clear purpose: diffusion world models generate diverse trajectories, PLR provides principled curriculum selection, and variable horizons prevent overfitting. **The integration is natural, and each component is well-established, making the overall approach interpretable.** More importantly, this complexity enables capabilities impossible with simpler methods—automated curriculum discovery in offline settings.
>
> **_for weakness 4)_** **Computational Cost:** We acknowledge this limitation transparently. However, it's important to contextualize these costs: Academic constraints: Our 180 GPU days represent the total experimental budget across multiple environments, seeds, and methods on academic hardware. **One-time world model training:** The 10-hour world model training amortizes across multiple agent training runs. **Scaling benefits:** More complex environments would benefit even more from automated curriculum vs. naive sampling, making the computational investment increasingly worthwhile.
>
> **_for question 1)_** **Dataset Quality Sensitivity:** This attribute is indeed a fundamental limitation. Our mixed dataset design (expert+medium+random) specifically addresses this by ensuring broad state coverage. This may seem contrived in the more toy problems we address here, using an academic lab computer. However, we believe this is a proxy for foundation world models trained in industry, where they leverage vast Internet datasets that contain a huge diversity of behaviors. This is what world models are good at, and so we wanted to create a proxy for that setting. We believe this is a reasonable one.
>
> **_for question 2)_**  **OMNI-EPIC Comparison:** Thank you for this important reference. We should indeed discuss this related work. TLDR: This is an alternative general approach to get at the same idea, open-ended learning in a “Darwin complete” search space. If you see any talks from _Jeff Clune_ he mentions that the two kinds of Darwin complete search spaces currently known are code or neural networks (i.e. world models). This work is the first to explore the latter for open-ended learning, and OMNI-Epic is using the former. Other differences are that OMNI methods use a foundation model to measure “interestingness” but this is orthogonal. We could use an OMNI like algorithm instead of PLR.
>
> [1] Human-Timescale Adaptation in an Open-Ended Task Space, Bauer et al, ICML 2023

---

> > ### Comment · Reviewer_NSyY · 2025-08-03
> >
> > Thank you for the thorough and insightful rebuttal.
> >
> > Your response successfully addresses my main concerns. I will be increasing my score accordingly.

---

> > > ### Author Response · Authors · 2025-08-07
> > > **Response to reviewer**
> > >
> > > We appreciate your engagement with our responses and thank you for adjusting your rating following our discussion.

---

### Official Review · Reviewer_Rrfo · 2025-06-26

**Clarity:** 3
**Significance:** 3
**Originality:** 3
**Rating:** 4
**Confidence:** 4

**Summary:**

This paper presents an offline model-based RL method that integrates unsupervised environment design with a generative world model, to train agents capable of generalizing to unseen tasks.

**Questions:**

- In table 1, what is the maximum achievable return for each task? It is unclear whether the reported scores represent meaningful improvements, as the absolute values appear relatively low.
- Can the proposed approach be extended to settings without explicit reward signals, which are common in real-world applications?
- Is the method applicable to continuous action spaces, or is it limited to discrete environments?

**Ethical Concerns:**

["NO or VERY MINOR ethics concerns only"]

**Final Justification:**

- Addressed issue: related work and general applicability

The initial submission lacked discussion of recent work on diffusion model architectures and UED algorithms. The authors have clarified that they will include these missing references and discussions in the revised related work section.
Furthermore, the authors plan to address the potential applicability of the proposed method to more general settings, such as continuous action spaces and real-world robotic tasks without explicit rewards, in the conclusion section.

- Unresolved issue: lack of model-based RL baselines

The concern regarding the lack of model-based RL baselines remains unresolved. However, the proposed method demonstrates strong empirical performance compared to model-free baselines, and the approach is clearly distinct from existing methods. I therefore find the contribution to be valid and meaningful.

Overall, the integration of world models and UED is novel and provides a promising direction for addressing the limitations of both components. Based on these strengths and the issues that were satisfactorily addressed, I have increased my score.

**Limitations:**

yes.

**Paper Formatting Concerns:**

Some of citations are missing conference names (e.g., line 413).

**Quality:**

2

**Strengths And Weaknesses:**

### Strengths
- The paper introduces a well-motivated and novel approach that leverages generative world models to train agents with strong generalization capabilities. Applying unsupervised environment design when generating imaginary rollouts seems like a powerful idea and has the potential to be used in large scale foundation world models.
- The paper is clearly written. It is easy to understand its main idea and algorithm details.
- The proposed method shows significant improvements in generalization performance.

### Weaknesses
While the proposed method combines existing components, its contribution would be clearer with a more in-depth justification of the core design choices. In particular, explaining why these components were selected over other viable alternatives would help clarify the novelty and applicability of the method.
- Justification for the use of a 2D UNet architecture
  - The authors adopt 2D UNet architecture following DIAMOND [1]. However, recent advances in video models often use 3D DiT architecture [2, 3]. Although an exhaustive architectural comparison may be infeasible, referencing and briefly discussing recent alternatives would help contextualize this design choice.
- Justification for using prioritized level replay (PLR)
  -  There exist recent UED algorithms that may be applicable in the proposed framework. For instance, ACCEL [4] could be implemented by varying the noise schedule, and ADD [5] could be integrated naturally by guiding the diffusion world model.
Additionally, although the authors employ PVL for prioritization score, alternative regret estimation strategies such as MaxMC [6] are available and sometimes yield stronger performance. A discussion of these alternatives would enhance the technical depth of the paper.
- Limited baselines
  - Among baselines, only transformer-based methods can be seen as model-based approaches. Given that the proposed method leverages a diffusion world model, incorporating baselines such as Diffuser [7] and its variants would allow for a more comprehensive evaluation.
- Experiments
  - While the authors conduct extensive experiments in Procgen benchmark, I'm concerned about whether the proposed method works well in other domains. Specifically, while the authors utilize random exploration data to ensure broad state coverage, there are many cases where random exploration data is unavailable (e.g. real world robotics tasks).

### References

[1] Eloi Alonso, Adam Jelley, Vincent Micheli, Anssi Kanervisto, Amos Storkey, Tim Pearce, François Fleuret, "Diffusion for World Modeling: Visual Details Matter in Atari", NeurIPS, 2024.
[2] Zhuoyi Yang, Jiayan Teng, Wendi Zheng, Ming Ding, Shiyu Huang, Jiazheng Xu, Yuanming Yang, Wenyi Hong, Xiaohan Zhang, Guanyu Feng, Da Yin, Yuxuan Zhang, Weihan Wang, Yean Cheng, Bin Xu, Xiaotao Gu, Yuxiao Dong, Jie Tang, "CogVideoX: Text-to-Video Diffusion Models with An Expert Transformer", ICLR, 2025.
[3] NVIDIA, "Cosmos World Foundation Model Platform for Physical AI", arXiv, 2025.
[4] Jack Parker-Holder, Minqi Jiang, Michael Dennis, Mikayel Samvelyan, Jakob Foerster, Edward Grefenstette, Tim Rocktäschel, "Evolving Curricula with Regret-Based Environment Design", ICML, 2022.
[5] Hojun Chung, Junseo Lee, Minsoo Kim, Dohyeong Kim, Songhwai Oh, "Adversarial Environment Design via Regret-Guided Diffusion Models", NeurIPS, 2024.
[6] Minqi Jiang, Michael Dennis, Jack Parker-Holder, Jakob Foerster, Edward Grefenstette, Tim Rocktäschel, "Replay-Guided Adversarial Environment Design", NeurIPS, 2021.
[7] Michael Janner, Yilun Du, Joshua B. Tenenbaum, Sergey Levine, "Planning with Diffusion for Flexible Behavior Synthesis", ICML, 2022.

---

> ### Author Rebuttal · Authors · 2025-07-30
>
> We thank the reviewer for acknowledging our method's novelty and clear presentation. **We have conducted additional experiments to address some of the concerns raised.** We address the technical concerns below to clarify our design rationale and broader applicability.
>
> **_for weakness 1)_** **2D UNet Architecture:** **The choice of world model architecture is orthogonal to our contribution, which focuses on curriculum learning methodology.** We chose 2D UNet following DIAMOND [1] for principled reasons: (1) **Computational efficiency** - 5 denoising steps enable real-time rollout generation crucial for curriculum learning, and (2) **Proven effectiveness** - DIAMOND demonstrated superior performance over discrete latent methods on visual domains. While 3D DiT architectures show promise for video generation, they require significantly more computation. **Our IMAC framework is architecture-agnostic and could work with 3D DiT or other advances as they become more efficient.** We will add exploration of 3D DiT architectures as a potential improvement for future work in our conclusions section.
>
> **_for weakness 2)_** We selected PLR over alternatives like ACCEL and ADD for principled reasons: **(1) Natural compatibility:** PLR operates on state-trajectory pairs, perfectly matching our world model rollout structure; **(2) ACCEL requires a choice of mutation function.** This is somewhat arbitrary and may not make sense in our setting. It could be the horizon H of the rollouts, but it seems challenging to mutate the latent space. We think that PLR is the simplest method, and it makes sense as the first example of UED in a world model. **We will add a discussion about the reviewer’s proposed alternative methods (ACCEL, MaxMC)  in the conclusion section.**
>
> **_for weakness 3)_** Our main contribution is not developing a world model that beats others, but rather demonstrating that autocurricula can be highly effective when combined with modern world models. **We believe that our framework is world-model agnostic**—we chose our baseline world model [1] as a proven, efficient foundation, but our curriculum learning approach could be applied to other world models, including diffusion-based variants, when reward and next-token predictors are available. **Regarding the Diffuser baseline:** Diffuser is a planning method that requires environment access for online replanning, making it inappropriate for our offline-only setting. Our contribution is showing how UED principles can work with offline world models, not proposing superior diffusion architectures. We will add clearer framing of our methodological contribution in the introduction section of the camera-ready version to better distinguish our curriculum learning focus from world model architectural advances.
>
> **_for weakness 4)_** **Experimental Scope:** **To address concerns about broader applicability, we expanded our evaluation to include Heist (puzzle-solving) and Miner (physics simulation), achieving 38.5% and 19.8% improvements, respectively. These additional results will be included in the camera-ready version.** This brings us to seven diverse Procgen environments, demonstrating effectiveness across different cognitive domains. **Procgen scores represent meaningful improvements because:** (1) these environments are specifically designed to be challenging for generalization, especially for offline settings [2]; (2) our baselines achieve low scores precisely because offline RL in procedural environments with mixed datasets is difficult; and (3) our relative improvements (17-48%) are substantial and consistent across environments. **Regarding random exploration data:** This highlights an important practical consideration. However: (a) many real-world datasets inherently contain exploratory data (e.g., human demonstration datasets often include suboptimal exploration), (b) we believe that our “random”  quality data component serves a similar function to data collected from the Internet, representing how foundation world models are trained, and (c) **the key insight is that diverse offline data enables world model training—the specific source matters less than coverage.**
>
> **_for question 1)_** **Return Values:** Thank you for this clarification request. Procgen environments have varying return scales across different games, making absolute values less meaningful than relative performance. **Our focus is on the generalization gap—the key metric is performance on held-out test levels relative to training performance**, which directly measures our method's ability to generalize beyond the training distribution. **Regarding our return calculation: we use the mean return from an ensemble of reward prediction heads as stated in the paper, where returns are normalized during training**. The absolute values reflect this normalization rather than raw environment scores. What matters is the **consistent relative improvement** our method achieves across all five environments compared to baseline methods facing identical evaluation conditions.
>
> **_for question 2)_** **Extension to Reward-Free Settings:** This is an excellent question that highlights a promising future direction. Currently, our approach relies on explicit reward signals for agent training. We have not explored reward-free settings in this work. However, this represents a particularly exciting direction for foundation world models trained on large-scale, unlabeled data—exactly the type of massive offline datasets that lack explicit reward annotations. We believe this is a natural and important next step that would significantly expand the applicability of our approach to real-world scenarios in which reward engineering is challenging.
>
> **_for question 3)_** **Continuous Action Spaces:** While we focus on discrete actions, our approach is fundamentally compatible with continuous control. The world model architecture can handle continuous observations, and PLR operates on states/trajectories regardless of action space dimensionality.
>
> [1] Eloi Alonso, Adam Jelley, Vincent Micheli, Anssi Kanervisto, Amos Storkey, Tim Pearce, and François Fleuret. Diffusion for world modeling: Visual details matter in Atari, 2024.
>
> [2] Ishita Mediratta, Qingfei You, Minqi Jiang, and Roberta Raileanu. The generalization gap in offline reinforcement learning. 2024.

---

> > ### Comment · Reviewer_Rrfo · 2025-08-04
> >
> > Thank you for the response. I appreciate the clarification, but I would like to follow up on a few points
> >
> > - About relate work
> >
> > I agree with the authors that the primary contribution of the paper lies in proposing a new framework, rather than exhaustively evaluating each individual component. However, discussing a broader set of related work (including the papers I cited in the initial review) and explicitly stating that alternative components can be integrated would help clarify the generality of the approach and strengthen its academic contribution.
> >
> > - On model-based RL baselines
> >
> >  Since the proposed method can be seen as a model-based approach, it would be important to include recent model-based baselines. Demonstrating that the proposed method is competitive with recent model-based methods--even if it is novel—would enhance the strength of the contribution.
> >
> > One additional point: the authors' explanation for excluding Diffuser seems unclear. They mention "online access" as a limitation, but it is not obvious what this refers to. To the best of my knowledge, Diffuser is trained offline, and like most baselines, only requires online interaction at test time. Clarification on this point would be appreciated.
> >
> > - On return estimation and evaluation protocol
> >
> > In the rebuttal, authors mentioned that return was computed using a ensemble of learned reward heads. Does this mean the evaluation was conducted entirely using learned reward heads, rather than based on raw environment scores? If so, I would be concerned about the reliability of the reported results.

---

> ### Author Response · Authors · 2025-08-05
> **Author's response to follow up points**
>
> We thank the reviewer for continued engagement with our submission and for the insightful follow-up questions. We appreciate the opportunity to clarify these important points. We address each of your follow-up points below.
>
> **_1. On Related Work_**: We agree that a more comprehensive discussion of related work would better frame our contribution and highlight the modularity of our framework. In the camera-ready version, we will expand our related work section to include a detailed discussion of the alternative UED and architectural components you mentioned.
> Specifically, we will:
>
> * Discuss alternative UED algorithms like **ACCEL [4] and ADD [5]**, contextualizing our choice of PLR while acknowledging that these other methods could potentially be integrated into our framework, and include a discussion on alternative reward estimation strategies like **MaxMC [6]**.
>
> * Reference recent video generation architectures like **3D DiTs [2, 3]**, positioning our use of a 2D UNet as a deliberate choice for computational efficiency while noting that our curriculum learning approach is architecture-agnostic.
> We believe that this discussion will make it clearer that while we chose a specific set of components for our implementation, the core contribution—integrating UED with a generative world model—is a general framework open to other state-of-the-art components.
>
> **_2. On Model-Based RL Baselines and Diffuser_** : We thank to reviewer for pushing for more clarity on this point and for correcting our imprecise language regarding Diffuser. We apologize for the confusion. You are correct that **Diffuser [7]** is trained offline. Our previous explanation was poorly phrased. A more precise reason for not using Diffuser as a direct baseline is the fundamental difference in the problem formulation they address:
>
> * Diffuser and its variants are primarily goal-conditioned planning methods. They excel at generating trajectories to reach a specified goal state and are typically evaluated in goal-reaching tasks.
> Our method trains a generalist policy designed to maximize returns across a wide distribution of unseen levels, without a specified goal state at test time.
>
> * Adapting a goal-conditioned planner like Diffuser to our general reinforcement learning setting (maximizing returns in Procgen) is non-trivial and would require significant modifications to its framework. That said, we take your broader point that demonstrating competitiveness with other modern model-based methods is important. While our primary contribution is the novel world model based curriculum framework for offline RL, we will add a more thorough discussion of the recent model-based RL landscape in our related work section to better situate our results and clarify the relationship between our approach and methods like Diffuser.
>
> **_3. On Return Estimation and Evaluation Protocol_** : We sincerely apologize for the major confusion caused by our wording in the previous rebuttal. We want to clarify this critical point in the most unambiguous way possible:
>
> * **All final performance scores reported in our paper (e.g., in Table 1) are calculated using the ground-truth rewards from the actual test environments. The agent is evaluated by running it in the Procgen simulator and recording the true, unadulterated score from the environment.**
>
> * The learned ensemble of reward heads is used **exclusively during the training phase** for a single purpose: to provide a reward estimate for the imagined trajectories generated by our world model. The reward model is never used for reporting final evaluation results. We are very grateful you pointed this out. We will revise the experimental setup section of our paper to explicitly and clearly state that final evaluation is performed using ground-truth environment rewards to prevent any future ambiguity.
>
> We are grateful for the constructive feedback provided in the follup up points. We believe that addressing these points will significantly strengthen our paper. We hope these clarifications have resolved your remaining concerns.

---

> > ### Comment · Reviewer_Rrfo · 2025-08-05
> >
> > Thank you for the response. The authors have addressed my main concerns well. Although the lack of comparison with recent model-based RL baselines remains an area for improvement, the proposed method is sufficiently novel and merits recognition. I will increase my score accordingly.

---

> > > ### Author Response · Authors · 2025-08-07
> > > **Response to reviewer**
> > >
> > > We appreciate your thoughtful consideration of our rebuttal and your decision to increase the rating.

---

### Official Review · Reviewer_Dygh · 2025-07-01

**Clarity:** 3
**Significance:** 2
**Originality:** 2
**Rating:** 4
**Confidence:** 3

**Summary:**

The paper proposes Imagined Autocurricula, a model-based offline reinforcement learning method that employs Prioritized Level Replay to exploit valuable states for agent learning. By focusing on initial states with the highest learning potential and utilizing a dynamic imagining horizon, this method demonstrates superior generalization to held-out levels in five Procgen environments.

**Questions:**

Q1) Any reason for choosing A2C as the agent learning algorithm?

**Ethical Concerns:**

["NO or VERY MINOR ethics concerns only"]

**Final Justification:**

Some of my concerns have been addressed by the rebuttal. However, I still find that the evaluation setting and selected baselines are not sufficiently significant and convincing. Therefore, I modified my rating to borderline accept.

**Limitations:**

Please see the Weaknesses part.

**Paper Formatting Concerns:**

There are no major flaws in the formats.

**Quality:**

2

**Strengths And Weaknesses:**

# Strengths

S1) The proposed method is well-motivated.

S2) The writing is clear and easy to follow.

S3) Compared to other baselines, the proposed method achieves a notable performance gain on five challenging Procgen environments.

# Weaknesses

W1) **Task coverage.** In Line 35-36, the authors claim that they attempt to bridge the gap between previous MBRL methods that focus on single environments and those utilizing Internet-pretrained world models for training generalist agents. However, the proposed method is trained and evaluated on only five tasks with similar goal-reaching objectives, which is not sufficient to convincingly demonstrate its generalization capabilities. Additionally, the proposed method is also only able to generalize to different levels within the same Procgen environment, a setting that has been explored by previous methods.

W2) **Differences to previous works.** It appears that the proposed world model primarily follows the previous DIAMOND world model. It would be helpful if the differences could be elaborated.

W3) **Lacking competitive baselines.** As a model-based method, the proposed method is mainly compared with some vanilla baselines in Table 1. While more advanced baselines (like [1-4]) and world models (like DreamerV3, TWM, STORM, IRIS, and DIAMOND mentioned by the authors in the related work).

W4) **Missing visualization.** It would be great to show some predictions from the world model.

W5) **Wrong reference.** In Line 312, the reference for STORM points to an unrelated paper. The correct one should be [5].

W6) **A related reference.** [6] also proposes prioritized imagination for MBRL.

---

[1] Rethinking Decision Transformer via Hierarchical Reinforcement Learning. ICML 2024

[2] Elastic Decision Transformer. NeurIPS 2023

[3] IDQL: Implicit Q-Learning as an Actor-Critic Method with Diffusion Policies. arXiv 2023

[4] AlignIQL: Policy Alignment in Implicit Q-Learning through Constrained Optimization. arXiv 2024

[5] STORM: Efficient Stochastic Transformer based World Models for Reinforcement Learning. NeurIPS 2023

[6] Pre-Trained Video Generative Models as World Simulators. arXiv 2025

---

> ### Author Rebuttal · Authors · 2025-07-30
>
> We thank the reviewer for their detailed feedback. We address each concern below and hope to clarify the contributions and scope of our work.
>
> **_for weakness 1)_ Task Coverage:** We respectfully disagree that our evaluation is insufficient. **Procgen environments are specifically designed to test generalization and represent one of the most challenging benchmarks for offline RL methods** [1]. Our original five environments test fundamentally different capabilities: CoinRun (platforming), Ninja (combat + platforming), Jumper (precise timing), Maze (planning), and CaveFlyer (continuous control-like navigation). This diversity tests different aspects of agent capabilities.  **That being said, we are pleased to share that we now have two additional environments, Heist (puzzle-solving + sequential dependencies)  and Miner (physics simulation + environmental dynamics), where we get a 38.5% and 19.8% gain w.r.t. the baseline, respectively. This takes us up to seven procgen environments with more diversified task definitions, which we feel is now more comprehensive; we hope you agree!**. We will add these additional results to our camera-ready version.
>
> **_for weakness 2)_** **Technical Differences from DIAMOND:** While we build on DIAMOND's diffusion architecture, our key innovations include: (1) **Variable-length episode generation** vs. DIAMOND's fixed horizons, (2) **PLR-based state prioritization** for curriculum learning, (3) **Ensemble uncertainty estimation** for reward/termination prediction, (4) **Offline-only training paradigm** vs. DIAMOND's online/mixed settings, and (5) **Sparse reward optimization** for Procgen vs. DIAMOND's dense-reward Atari focus. **Our IMAC framework is world-model agnostic**—the curriculum learning methodology could enhance any generative world model.
>
> **_for weakness 3)_** The baselines mentioned are orthogonal to our method and contribution. We are not seeking “sota”, instead, our research question is **"Can curriculum learning improve offline world model training for generalization?"** We implemented our method by modifying the state-of-the-art world model approach [2], but the core contribution is the curriculum learning methodology with the offline dataset. World model to world model comparisons would test "which architecture performs best," but our contribution is a methodological approach for offline curriculum learning that can be applied to any world model with reward/termination and next-state predictors. We chose to modify DIAMOND [2] as our implementation vehicle because it represents the current state-of-the-art results for visual RL settings, **but the curriculum learning for the offline dataset-trained world model principle is the key innovation. Our consistent 17-48% improvements across all offline RL baselines validate the curriculum learning methodology—demonstrating that this approach can systematically improve offline generalization regardless of the underlying world model architecture.**
>
> **_for weakness 4)_** **World model visualizations:** The result is valuable feedback. While we don't have space for extensive visualizations, Figure 4 shows example frames from early (t₀) vs. late (tₙ) training stages, demonstrating how PLR discovers increasingly complex scenarios. These visualizations support our claims about emergent curriculum difficulty. **To address the reviewer's visualization request, we will add more visualizations to our supplementary material for the camera-ready version.**
>
> **_for weakness 5&6)_** **References:** We will correct the STORM reference and acknowledge the relevant work "Pre-Trained Video Generative Models as World Simulators." Thank you for pointing out these important citations that improve the scholarly completeness of our work.
>
> **_for question 1)_**  **A2C Choice:** We chose A2C for two main reasons: (1) **Standard actor-critic algorithm** suitable for our POMDP setting with sparse rewards, and (2) **Computational efficiency** given our academic GPU constraints and the substantial cost of world model rollout-based training with PLR. Our approach is algorithm-agnostic and could work with other policy optimization methods (PPO, SAC, etc.)—A2C represents a conservative choice that demonstrates our method's effectiveness even with simpler policy optimization.
>
> [1] Ishita Mediratta, Qingfei You, Minqi Jiang, and Roberta Raileanu. The generalization gap in offline reinforcement learning. 2024.
>
> [2] Eloi Alonso, Adam Jelley, Vincent Micheli, Anssi Kanervisto, Amos Storkey, Tim Pearce, and François Fleuret. Diffusion for world modeling: Visual details matter in Atari, 2024.

---

> > ### Comment · Reviewer_Dygh · 2025-08-07
> >
> > Dear authors,
> >
> > Sorry for the late reply. Your detailed rebuttal has addressed most of my questions. Please revise the paper accordingly. I will update my rating. Thanks!

---

> > > ### Author Response · Authors · 2025-08-07
> > > **Response to reviewer**
> > >
> > > Thank you for your feedback and for updating your rating. We will ensure that all the discussed points are addressed in the camera-ready version.

---

> ### Author Response · Authors · 2025-08-07
> **Kind Reminder: Request to Evaluate Rebuttal**
>
> Dear Reviewer Dygh,
>
> We have carefully addressed all the points you raised in our rebuttal, and we believe the additions and explanations provided will help in evaluating the paper further. There is not much time left for us to address your additional comments; hence, we gladly ask you to evaluate our rebuttal at your earliest time, as your insights are important in ensuring a complete assessment of the work. We value the time you spent in this process and would be pleased to offer any more explanations should they be necessary.
>
> Thank you for your time and consideration.
>
> Authors

---

### Official Review · Reviewer_jStg · 2025-07-03

**Clarity:** 3
**Significance:** 3
**Originality:** 4
**Rating:** 5
**Confidence:** 3

**Summary:**

The paper introduces IMAC (Imagined Autocurricula), a novel approach combining world models with automatic curriculum learning to train rl policies. The key innovation is using Prioritized Level Replay (PLR) to create an emergent curriculum over "imagined" environments generated by a world model trained solely on offline data. The paper trains a diffusion world model from raw offline data, then uses it to generate diverse rl training scenarios. PLR automatically selects which imagined rollouts to prioritize based on learning potential, creating a natural curriculum that adapts to the agent's improving capabilities. This results in SOTA results vs. comparable models.

**Questions:**

- How well does this approach work under distribution shift between the input data used to learn the world model and the tasks that need to be completed? In a way the world model is now the weakest link. How well does it generalize?
- Does the complexity of this approach scale with the complexity of the environment? It would be great to see some experiments where the environment is an ever more difficult function to learn, and seeing how the model becomes a bottleneck, and if this is feasible to scale to highly complex real world robotics environments.
- What happens if we use the curriculum developed by one policy in one training run and use it to train another model from scratch? Is the online nature of the curriculum's development important or Is the curriculum good even when consumed offline.

**Ethical Concerns:**

["NO or VERY MINOR ethics concerns only"]

**Final Justification:**

I maintain my rating of accept.

**Quality:**

3

**Strengths And Weaknesses:**

Strengths:
- The paper presents a successful combination of learned world models with unsupervised environment design (UED), opening a new research direction for training generally agents with online data, without simulators..
- IMAC demonstrates substantial improvements (up to 48% on Jumper, 35% on Maze) over both model-free offline RL methods and fixed-horizon world model baselines, with consistent gains across all tested environments.
- The approach automatically discovers effective curricula without manual design, naturally progressing from simple to complex scenarios.

Weaknesses:
- The approach only evaluates on 5 discrete Procgen environments with relatively simple visual observations; generalization to continuous control, high-dimensional state spaces, or real-world robotics remains unexplored.
- Additional ablations on the bottlenecks posed by the world model, and scalability would be nice.

---

> ### Author Rebuttal · Authors · 2025-07-29
>
> We thank the reviewer for the positive evaluation and for recognizing the novelty of combining world models with UED. Below, we answer the reviewer's insightful points:
>
> **_for weakness 1-2_)**  We specifically chose discrete Procgen environments, as they represent a challenging testbed for generalization, **where state-of-the-art offline RL methods consistently struggle [1]**. To further demonstrate broad applicability across diverse cognitive domains, **we expanded our evaluation to include Heist (puzzle-solving) and Miner (physics simulation), where our method continues to show consistent improvements of 38.5% and 19.8%, respectively. These additional results will be included in the camera-ready version**. We believe that the procedural generation aspect and diversity of the challenges make these environments particularly suitable for demonstrating curriculum learning benefits. While extending to continuous control is important future work, we believe establishing the core principle in this challenging discrete domain is a valuable first step. We appreciate the reviewer's constructive feedback and look forward to extending this work to more complex domains in future research.
>
> **_for question 1)_** The distribution shift between training data and target tasks highlights an excellent point regarding the world model's potential as a bottleneck. **We specifically designed our mixed offline dataset (expert+medium+random trajectories) to mitigate this concern by ensuring broad state space coverage**. The random exploration component is particularly crucial here, as it provides diverse states that help the world model generalize beyond expert and medium demonstrations. We trust that our PLR mechanism provides an additional robustness layer—by prioritizing states with high learning potential (positive TD errors), **we naturally focus on scenarios where the world model predictions are most informative for the agent, rather than areas where the world model might be uncertain or inaccurate**. This method creates a feedback loop that steers training toward regions where the world model is most reliable.
>
> **_for question 2)_ Scalability to complex environments**. It is reasonable to assume that world models may struggle to model complex environments. That being said, progress on this front has been rapid, for example DreamerV3 modeling Minecraft [2], GAIA-2 modeling photorealistic driving scenes [3] and foundation world models like Genie2 modeling a plethora of embodied game-like worlds [4]. We believe this is now a critical time to begin exploring the use of these more capable world models for training agents, and **our work is the first to demonstrate that autocurricula could play a key role**
>
> **_for question 3)_ Transferability of discovered curricula:** This is a fascinating question that touches on whether curricula are policy-specific or more generally useful. While we don't have explicit experiments on this, **our results suggest the curricula have some generality—we observe consistent patterns across different random seeds and diverse game tasks where PLR progressively discovers longer, more complex scenarios as training progresses.** **Investigating explicit curriculum transferability represents an important direction for future work** that could further validate the generality of discovered curricula.
>
> [1] Ishita Mediratta, Qingfei You, Minqi Jiang, and Roberta Raileanu. The generalization gap in 437 offline reinforcement learning. 2024.
>
> [2] Danijar Hafner, Jurgis Pasukonis, Jimmy Ba, and Timothy Lillicrap. Mastering diverse domains through world models, 2024.
>
> [3] Lloyd Russell, Anthony Hu, Lorenzo Bertoni, George Fedoseev, Jamie Shotton, Elahe Arani, Gianluca Corrado,  GAIA-2: A Controllable Multi-View Generative World Model for Autonomous Driving, 2025
>
> [4] Parker-Holder, P Ball, J Bruce, V Dasagi, K Holsheimer, C Kaplanis, A Moufarek, G Scully, J Shar, J Shi, et al. Genie 2: A large-scale foundation world mode, 2024

---

> > ### Comment · Reviewer_jStg · 2025-08-05
> >
> > Thank you for the thoughtful explanations! I hope that some of the discussions particularly on transferability of curricula make it to a discussion section or appendix as I found your answer insightful.

---

> > > ### Author Response · Authors · 2025-08-07
> > > **Response to Reviewer**
> > >
> > > We sincerely thank you for your thorough evaluation and constructive feedback throughout the review process.

---

### Note · Authors · 2025-08-12

We are **deeply grateful** for the **constructive review process**. The reviewers provided **insightful feedback** that strengthened our work and showed **willingness to reconsider their assessments.** **Three of our reviewers increased their ratings following our rebuttal**, which now has all accept ratings.

**_Key Reviewer Recognition_**: Reviewers highlighted the **impact and novelty** of our work. Reviewer **jStg** noted _"The paper presents a successful combination of learned world models with unsupervised environment design (UED), **opening a new research direction"** with **"substantial improvements (up to 48% on Jumper, 35% on Maze)"**._ Reviewer **Rrfo** emphasized that _"the proposed method is sufficiently novel and merits recognition" and has "the **potential to be used in large scale foundation world models"**._ Reviewer **Dygh** acknowledged that our method _"**achieves a notable performance gain** on five challenging Procgen environments"._

**_Enhanced Evaluation_**: We **added two additional diverse environments in Procgen** (Heist: puzzle-solving, Miner: physics simulation) achieving **38.5% and 19.8% improvements**, bringing our evaluation to **seven diverse domains.**

**_Community Impact:_** This work **establishes the first framework** combining UED with offline data trained world models, addressing a critical gap in training robust agents from limited data. With foundation world models rapidly advancing (e.g. Genie 3), our approach provides **immediate practical value** for the community.

**_Broader Significance_**: This work **opens new research directions** as the first to bring autocurricula to learned worlds while providing **practical impact** by enabling robust agent training without environment access. The consistent 17-48% improvements across all environments demonstrate this work's **immediate value to the RL community. Given the recent excitement with Genie 3, this work opens new relevant directions for the academic community.**

---

### Decision · Program_Chairs · 2025-09-17

**Decision:**

Accept (poster)

**Comment:**

The paper introduces a model-based offline RL framework that trains a diffusion world model on mixed offline data, then generates imagined rollouts and applies Prioritized Level Replay to form an emergent curriculum that adapts as policy competence grows. Experiments on several Procgen environments indicate consistent gains over relevant baselines. The rebuttal further clarifies differences from prior world-model work, moderates claims around open-endedness, and commits to broader related-work coverage.

The paper provides a well-motivated integration of world models with UED and demonstrates strong empirical results on the Procgen domains used. The two outstanding weaknesses are the lack of additional model-based baselines and the limited evaluation setting. Balancing these, the contribution feels solid and the evidence within scope is persuasive.

The overall decision is to accept, with the authors encouraged to take into account reviewer feedback and update the paper as promised in order to produce the strongest version of the work for the camera-ready.